# Alleviating Soil Acidification and Increasing the Organic Carbon Pool by Long-Term Organic Fertilizer on Tobacco Planting Soil

**Peigang Dai** [1,2,†], **Ping Cong** [2,†], **Peng Wang** [2], **Jianxin Dong** [2], **Zhaorong Dong** [1,*] **and Wenjing Song** [2]

[1]  College of Agronomy, Anhui Agricultural University, Hefei 230036, China; daipeigang@caas.cn
[2]  Tobacco Research Institute of Chinese Academy of Agricultural Sciences, Qingdao 266101, China; congping@caas.cn (P.C.); wacion.qi@gmail.com (P.W.); dongjianxin@caas.cn (J.D.); songwenjing@caas.cn (W.S.)
[*]  Correspondence: dongzhaorong@ahau.edu.cn; Tel.: +86-0551-657-862-13
[†]  Contributed equally.

**Abstract:** Long term tobacco planting leads to soil acidification. A ten-year experiment with various fertilization treatments (no fertilization (CK), chemical fertilizer (CF), organic-inorganic compound fertilizer (OCF), and organic fertilizer (OF)) was carried out between 2010 and 2020 in a continuous cropping system of *Nicotiana tabacum* in the brown soil of eastern China, to assess the effects of organic fertilizer on the improvement of tobacco planting soil acidification. The results indicated that treatments OCF and OF reduced the soil exchangeable acid content, of which the exchangeable aluminum showed the largest reduction by 51.28% with the OF treatment. In contrast, treatment CF showed more significant increases in exchangeable aluminum（Al）and Al saturation, and also apparently increased soil $NO_3^-$-N, $NH_4^+$-N and nitrification potential (NP) than other treatments. Treatments of OCF and OF significantly increased the total amount of exchangeable base (EBC) by 37.19% and 42.00% compared with CF, respectively. Redundancy analysis (RDA) showed that NP, $NH_4^+$-N, and $NO_3^-$-N were the important factors indicating soil acidification, while EBC and exchangeable K were the significant factors restricting soil acidification. Inevitably, OCF could improve the soil organic carbon pool and labile organic carbon pool. The structural equation model (SEM) showed that OCF treatment increased the soil organic carbon pool mainly by inhibiting soil nitrification and reducing the content of exchangeable Al. In conclusion, both OF and OCF treatments were effective methods to alleviate tobacco planting soil acidification, however OCF had more advantages in improving soil organic carbon pool.

**Keywords:** organic fertilizer; soil acidification; organic carbon; tobacco

## 1. Introduction

Soil acidification is one of the most important factors limiting nutrient uptake and crop yields. Unlike many other crops, tobacco is a perennial leaf-harvested crop that grows best in an acidic soil, with an optimum pH between 5.5 and 6.5 [1,2]. However, an investigation of soil pH in seven tobacco planting areas in the Chongqing Municipal City, southwest China found that the pH value decreased by 0.2 units after ten-years of tobacco planting [2]. Soil acidification can lead to the accumulation of aluminum (Al) and deficiencies in phosphorus (P), potassium (K) and other nutrients in tobacco field soil [3]. Concerning the quality of tobacco, soil acidification can reduce the total sugar, protein, and aroma substances in tobacco leaves, increase metals ions such as aluminum, iron, and manganese, and is not conducive to the growth of tobacco or the formation of high-quality tobacco leaves [4]. Therefore, it is urgent to solve the problem of soil acidification in

tobacco fields, so as to ensure the further improvement of soil productivity and tobacco quality.

Generally, the long-term application of chemical fertilizer is an significant cause of soil acidification. Excessive nitrogen application especially is a common problem, and not only reduces the efficiency of nutrient utilization, but also causes potential environmental problems [5,6]. Specifically, it is thought that the accelerated soil acidification from N fertilization is directly caused by the production of protons via the nitrification process after ammonium nitrogen fertilization occurs [7,8]. The generated hydrogen ions ($H^+$) are buffered by a suite of factors, including carbonate, silicate, and exchangeable base cations, which depend on the soil pH [9]. However, as the alkaline cations are leached into the soil solution, the buffering capacity of the soil decreases under the increasingly acidic conditions [3,10].

Combined applications of organic and inorganic fertilizer are beneficial to abate soil acidification, improve soil properties [11,12] and maintain soil productivity [13]. The combined long-term application of organic fertilizer and inorganic fertilizer reduced exchangeable acid and aluminum content, while significantly increasing soil carbon and nitrogen storage capacity, as well as crop yield [14,15]. Organic fertilizer primarily contributes active organic carbon when it increases the total soil organic carbon (SOC) content [12]. Therefore, compared with chemical fertilizers, organic fertilizer is more conducive to soil organic carbon and nitrogen accumulation [16,17], which means the application of organic fertilizer changes the transformation processes of organic and inorganic nitrogen and reduces the $H^+$ produced by nitrification and the toxic effects of aluminum, so as to slow down the process of soil acidification. Two plausible reasons could explain this phenomenon. Firstly, the decomposition rate of animal manure or plant materials returned to the field is generally less than a half, which is lower than that of chemical fertilizer and will reduce the soil $H^+$ production [18]; Secondly, the organic acids produced by the compost can form complexes with the exchangeable aluminum in soil [19].

However, studies on the application of organic fertilizer to reduce soil acidity have been mostly concentrated in the southern red soil double-cropping rice planting area [8,20] of China. Brown soil is the most important type of tobacco planting soil in the Huang-Huai tobacco growing area, which is mainly distributed in the Jiaodong Peninsula and the hilly area of south-central Shandong Province. The annual planting area of tobacco in Shandong Province is more than 100,000 ha, which accounts for 29.20% of its cultivated land [21]. Nevertheless, this large area of tobacco growing soil is also threatened by soil acidification, which eventually leads to soil hardening, nutrient imbalance and production capacity reduction. Thus, we set up a long-term field experiment in the Shandong typical brown soil area beginning in 2010, including four treatments: no fertilization (CK), chemical fertilizer (CF), organic-inorganic compound fertilizer (OCF), and organic fertilizer (OF). In this study we hypothesized that the long-term application of organic fertilizer in a tobacco field could alleviate soil acidification, improve soil pH, and reduce exchangeable acid content and aluminum saturation. The relationship between exchangeable cations, nitrification and soil organic carbon with soil acidification will be further clarified. The objectives of this study were to investigate: (1) the effect of the long-term application of inorganic and organic fertilizers on soil pH, Ec, soil exchangeable acid, total exchangeable base, and ECEC; (2) the effects on soil cation exchange capacity and nitrification, and their relationships with soil acidification and (3) the response and causes of organic carbon pool in the process of fertilization regulating soil acidification.

## 2. Materials and Methods

### 2.1. Site Description

The long-term field experiment was established in the Qingdao tobacco resources and environment field scientific observation and Experiment Station of the Chinese Academy of Agricultural Sciences in 2010. It is located in Shimen village, Longquan street, Jimo District, Qingdao City, Shandong Province (36°26′54″ N, 120°34′38″ E; altitude = 75 m a.s.l.). The average precipitation is 708.9 mm, and the area has a temperate monsoon climate. The annual average temperature is 12.1 °C, the annual accumulated temperature is 4410 °C and the frost-free period is about 200 d. The local soil type is Mottlic Molli-Boric Argosols (Chinese Soil Taxonomy) with a sandy loam texture of 53% sand (0.05–1 mm), 24% silt (0.001–0.05 mm) and 4% clay (<0.001 mm) under the Chinese soil texture classification standard. The initial soil pH was 5.56 with the potentiometric method (water: soil = 2.5:1), soil bulk density was 1.20 g cm$^{-3}$ with the cutting ring method, SOC was 6.76 g kg$^{-1}$ with the potassium dichromate volumetric method, soil alkali hydrolyzed nitrogen was 52.69 mg kg$^{-1}$ with the alkaline hydrolysis diffusion method, soil available phosphorus was 10.60 mg kg$^{-1}$ with the extraction-molybdenum antimony colorimetric method, and soil available potassium was 105.25 mg kg$^{-1}$ with the flame spectrum method. All the basic physical and chemical properties of the cultivated soil layer (0–20 cm) were determined in May 2010.

### 2.2. Experimental Design

The experiment was carried out for ten years, from 2010 to 2020. Four treatments were included to evaluate the effects of the organic fertilizer on alleviating soil acidification:(i) no fertilization (CK); (ii) single application of chemical fertilizer (CF); (iii) application of organic fertilizer on the basis of inorganic fertilizer (OCF); and (iv) single application of organic fertilizer (OF), the organic fertilizer was made from cattle manure. The experiment used a randomized block design with three replicates. Each plot was 5.0 m × 4.4 m × 1.1 m and 40 tobacco plants were planted in a block. The tested *Nicotiana tabacum* variety was NC89, which was transplanted in the first ten days of June every year. The fertilizer was divided into base fertilizer, seedling fertilizer and top dressing. Applied chemical fertilizers were compound fertilizer (15-15-15) (SACF), potassium sulfate (0-0-50) (SDIC), diammonium phosphate (16-40-0) (YUNTIANHUA, China) and potassium nitrate (13-0-46) (XINRUNDE, China). The organic fertilizer was cow manure, containing 11.40 g kg$^{-1}$ N, 510.15 g kg$^{-1}$ P$_2$O$_5$, 19.55 g kg$^{-1}$ K$_2$O, 25.20 g kg$^{-1}$ calcium, 16.00 g kg$^{-1}$ magnesium, 201.27 g kg$^{-1}$ organic carbon. The specific fertilization amount used in each treatment is shown in Table 1. The field management was carried out according to the normal cultivation specifications of the farm. All fertilizers were applied to the surface in a certain amount, and then mixing with 0–10 cm soil manually, ridging and planting tobacco. The fertilizers were applied before tobacco planting every year.

**Table 1.** Nutrient application rate of different fertilization.

| Treatments | N (kg ha$^{-1}$) | | P$_2$O$_5$ (kg ha$^{-1}$) | | K$_2$O (kg ha$^{-1}$) | | Organic Carbon (kg ha$^{-1}$) | |
| --- | --- | --- | --- | --- | --- | --- | --- | --- |
| | Chemical Fertilizer | Organic Fertilizer | Chemical Fertilizer | Organic Fertilizer | Chemical Fertilizer | Organic Fertilizer | Chemical Fertilizer | Organic Fertilizer |
| CK | 0 | 0 | 0 | 0 | 0 | 0 | 0 | 0 |
| CF | 82.20 | 0 | 83.25 | 0 | 249.75 | 0 | 0 | 0 |
| OCF | 82.20 | 165.00 | 83.25 | 167.25 | 249.75 | 384.00 | 0 | 3019.05 |
| OF | 0 | 247.50 | 0 | 250.87 | 0 | 576.00 | 0 | 4528.57 |

Note: CK, no fertilization; CF, single application of chemical fertilizer; OCF, application of organic-inorganic fertilizer; OF, single application of organic fertilizer.

### 2.3. Sampling and Measurement

Soil samples were collected from each of the four incorporated treatments in the mature period of *Nicotiana tabacum* in September 2020. The sampled soil depth was 0–20 cm. The soil samples were collected from three points in each plot replicate and mixed to produce a composite sample. Each composite sample was sieved through a 2-mm mesh to remove plant tissues, roots, rocks, etc. [22] in the field. Then, it was taken back to the laboratory to determine the soil organic carbon, active organic carbon, pH, electric conductivity (Ec), exchangeable acid, exchangeable cation and other indicators.

The soil pH and electric conductivity (Ec) were determined at a soil: water ratio of 1:5 with a conductivity meter (FE38-FiveEasyPlus™, Mettler-Toledo, Zurich, Switzerland).

Soil exchangeable acid and exchangeable hydrogen was determined by 1 mol $L^{-1}$ potassium chloride exchange neutralization titration. Soil exchangeable calcium and magnesium were determined by 1 mol $L^{-1}$ ammonium acetate exchange atomic absorption spectrophotometry. Soil exchangeable potassium and sodium were determined by 1 mol $L^{-1}$ ammonium acetate exchange flame spectrophotometry [23].

The $NH_4^+$-N and $NO_3^-$-N in the soil were leached with KCl solution (1 mol $L^{-1}$), and the filtrate was taken and measured by a continuous flow analyzer (Seal-AA3, Germany). The determination of soil nitrification potential (NP) referred to the method of Taylor et al. [24] and Kandeler et al. [25]. Phosphate buffer (NaCl, 8.0 g $L^{-1}$; KCl, 0.2 g $L^{-1}$; $Na_2HPO_4$, 0.2 g $L^{-1}$; $NaH_2PO_4$, 0.2 g $L^{-1}$; pH 7.1) containing 1 mmol $L^{-1}$ $(NH_4)_2SO_4$ was added to fresh soil at a soil water ratio of 1:4. Then it was placed on a shaking table at 300 r $min^{-1}$ for dark culture at 25 °C. The soil suspension was cultured for 3, 6, 12 and 24 h, and then 2 mol $L^{-1}$ KCl extraction solution was added according to the soil water ratio of 1:10. After shaking at 180 r $min^{-1}$ for half an hour and filtration by suction, a flow analyzer was used to determine the nitrate nitrogen content in the filtrate. The linear relationship was fitted with the culture time (h) as the abscissa and the nitrate concentration (mg $kg^{-1}$) as the ordinate. The slope of the fitting equation, i.e., the growth rate of nitrate nitrogen content, was the soil nitrification potential value (mg $kg^{-1}$ $h^{-1}$).

The total organic carbon of the soil was determined by an HT1300 analyzer (Jena Analytik, Germany). The labile organic carbon was determined by the 1:0.2 $K_2Cr_2O_7$-$H_2SO_4$ method and 333 mmol $L^{-1}$ $KMnO_4$ oxidation method [26].

### 2.4. Calculations and Statistical Analysis

Exchangeable aluminum (Al), total exchangeable base content (EBC), and effective cation exchange capacity (ECEC) were calculated by the following formulas:

$$Q^+, Al^{3+} = Q^+, A - Q^+, H^+$$

$$Q^+, B = Q^+, K^+ + Q^+, Na^+ + Q^+, Ca^{2+} + Q^+, Mg^{2+}$$

$$Q^+, C = Q^+, K^+ + Q^+, Na^+ + Q^+, Ca^{2+} + Q^+, Mg^{2+} + Q^+, Al^{3+}$$

where $Q^+, Al^{3+}$ represents exchangeable Al; "$Q^+, A$" represents exchangeable acid; "$Q^+, H^+$" represents exchangeable hydrogen; "$Q^+, B$" represents total exchangeable base content; "$Q^+, C$" represents effective cation exchange capacity; "$Q^+, K^+$" represents exchangeable potassium (K); "$Q^+, Na^+$" represents exchangeable sodium (Na); "$Q^+, Ca^{2+}$" represents exchangeable calcium (Ca); and "$Q^+, Mg^{2+}$" represents exchangeable magnesium (Mg).

The carbon pool management index (CPMI) is an important index for representing the change of soil carbon pool, and was calculated as follows [27,28]:

$$CPMI = CPI \times AI \times 100$$

where CPI is the carbon pool index and AI is the activity index of the carbon pool. CPI and AI were calculated as follows:

$$CPI = \frac{TOCe}{TOCc}$$

$$AI = \frac{A_e}{A_c}$$

$$A = \frac{LOC(mg/g)}{NLOC(mg/g)}$$

$TOC_e$ represents the total organic carbon of the experimental soil; and $TOC_c$ represents the total organic carbon of the control soils. Here, we use abandoned land to represent control soil and the $TOC_c$ is 5.26 g $kg^{-1}$. $A_e$ is the carbon pool activity of the experimental soils; $A_c$ is the carbon pool activity of the control soils, and the $LOC_c$ is 0.79 g $kg^{-1}$. NLOC stands for the content of the non-labile organic carbon, which was the estimated as the difference between the TOC and LOC.

The analysis of the mean values and standard errors of all data were made in Excel 2016 (Microsoft, Redmond, WA, USA). One-way analysis of variance (ANOVA) of soil pH, Ec, soil exchangeable acid, EBC, ECEC, CPMI and other indicators were performed using SAS 9.4 (SAS, Cary, NC, USA). In one-way ANOVA, the means of treatments were compared using the least significant difference (LSD) at $p < 0.05$. The Pearson Correlation analysis was performed with R software package ('*Hmisc*') to determine whether there was a significant difference between soil acidification indexes. Redundancy analysis (RDA) was performed to investigate relationships between soil ions and soil acidification indexes by using CANOCO5 (CANOCO, Microcomputer Power Inc., Ithaca, NY, USA). Structural equation modeling (SEM) was performed by SPSS-AMOS to analyze hypothetical pathways to explain the impact of submergence on soil carbon pool management. The model fit was assessed by a $\chi^2$-test, the comparative fit index (CFI) and the root square mean error of approximation (RMSEA). Mean values ± SE are reported here.

### 3. Results

*3.1. Changes of Soil PH over Ten Years*

The trend of pH variation in tobacco planting soil from 2010 to 2020 have been shown in Figure 1a. After ten years of continuous fertilization, it was found that the changes of OF, OCF, and CK treatments were relatively minor, and the soil pH value of the OF treatment was the highest in each year. The application of organic fertilizer increased soil pH by 0.18 units in 2020 compared with 2010, while the application of chemical fertilizer alone led to a sharp decrease in soil pH by 0.84 units.

Compared with CK, the decrease in the soil pH value with CF treatment was the largest, ranging from 0.10 to 1.01 units during 2010 to 2020, and it showed the highest decrease in 2016.The pH value of the OF treatment was higher than that of the CK treatment during 2010 to 2020, and the degree of increase was 0.10 to 0.38 units. This indicated that long-term OF treatment could increase soil pH, while long-term CF treatment could decrease soil pH.

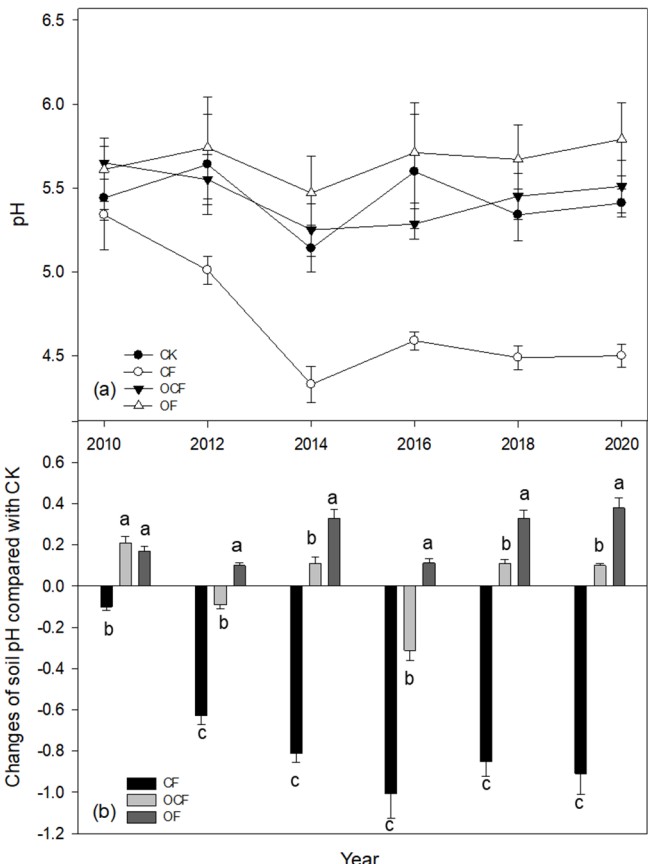

**Figure 1.** Trend of soil pH during the ten years (**a**) and changes of soil pH compared with CK (**b**) under different fertilization treatments. CK, no fertilization; CF, single application of chemical fertilizer; OCF, application of organic-inorganic fertilizer; OF, single application of organic fertilizer. Values with the same lowercase letter are not significantly different among the different fertilization treatments in the same year.

### 3.2. Soil PH, Ec and Soil Exchangeable Acid

The effects of different fertilization treatments on the soil pH, Ec and soil exchangeable acid were shown in Table 2. Compared with CK, CF significantly reduced the soil pH by 0.91 units ($p < 0.05$). The application of chemical fertilizer all the year round aggravated the acidification of tobacco planting soil. However, the OF treatment increased soil pH significantly, which was 0.36 units ($p < 0.05$) higher than CK. Long-term application of organic fertilizer improved the pH of soil, alleviating the acidification. Compared with CK, the exchangeable acid increased significantly with the CF treatment, of which the exchangeable Al increased significantly by 40.62% ($p < 0.05$). Compared with CK, OCF and OF significantly reduced the content of soil exchangeable acid, while OF had the highest decrease which was 37.50% ($p < 0.05$). These results showed that the application of organic fertilizer could reduce the exchangeable acid in soil, mainly as the exchangeable Al.

**Table 2.** Changes in pH, Ec and soil exchangeable acid under different fertilization treatments.

| Treatments | pH | Ec | Exchangeable Acidity (mmol kg⁻¹) | Exchangeable H (mmol kg⁻¹) | Exchangeable Al (mmol kg⁻¹) | Al Saturatio (%) |
|---|---|---|---|---|---|---|
| CK | 5.41 ± 0.04 b | 62.27 ± 0.74 b | 9.10 ± 0.10 a | 2.30 ± 0.07 a | 6.40 ± 0.04 b | 5.29 ± 0.89 bc |
| CF | 4.50 ± 0.12 c | 70.03 ± 0.51 a | 10.90 ± 0.10 a | 2.30 ± 0.03 a | 9.00 ± 0.06 a | 9.17 ± 0.21 a |
| OCF | 5.51 ± 0.08 ab | 60.63 ± 0.49 b | 6.70 ± 0.07 b | 2.00 ± 0.06 a | 5.40 ± 0.04 bc | 6.56 ± 0.66 b |
| OF | 5.79 ± 0.11 a | 55.40 ± 0.59 c | 5.10 ± 0.07 b | 2.00 ± 0.06 a | 4.00 ± 0.07 c | 4.36 ± 0.50 c |

Note: CK, no fertilization; CF, single application of chemical fertilizer; OCF, application of organic-inorganic fertilizer; OF, single application of organic fertilizer. Ec, electric conductivity. Different low case letters each line indicate significant differences ($p < 0.05$) among treatments using the least significant difference (LSD).

Correlation analysis showed that there was a significant negative correlation between pH and Ec ($p < 0.01$) in tobacco brown soil (Figure 2), indicating that the increase in pH led to the decrease in electrical conductivity. In addition, soil pH was significantly negatively correlated with exchangeable acidity, exchangeable Al and Al saturation ($p < 0.01$), indicating that the increase of exchangeable acidity, exchangeable Al and Al saturation was an important reason for the decrease in soil pH. Therefore, measures should be taken to reduce soil exchangeable acidity and exchangeable Al in order to alleviate the acidification of tobacco brown soil.

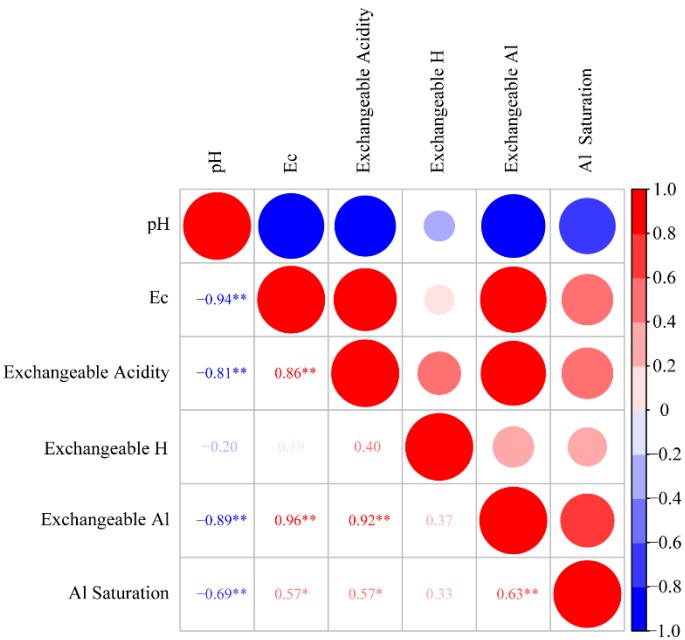

**Figure 2.** Correlation analysis among pH, Ec and soil exchangeable acid. The darker the color of the marker, the stronger the correlation. ** means $p < 0.01$, * means $p < 0.05$.

### 3.3. Soil Cation Exchange Capacity

The effects of different fertilization treatments on exchangeable K⁺, Na⁺, Ca²⁺ and Mg²⁺ in tobacco planting soil are shown in Table 3. Compared with CK, the different treatments had no significant effects on exchangeable K⁺, Na⁺, Ca²⁺ and Mg²⁺, even though the OCF and OF treatments increased the contents of exchangeable K⁺, Na⁺, Ca²⁺ and Mg²⁺.

Compared with CK, the OCF and OF treatment significantly increased the total exchangeable base by 15.93% and 20.00% ($p < 0.05$), respectively. This made the total amount of exchangeable base of the OCF and OF treatments significantly higher than that of CF by 37.19% ($p < 0.05$) and 42.00% ($p < 0.05$), respectively. However, none of the treatments had significant effects on the effective cation exchange capacity (ECEC).

**Table 3.** Changes in exchangeable $K^+$, exchangeable $Na^+$, exchangeable $Ca^{2+}$, exchangeable $Mg^{2+}$, exchangeable base cations and ECEC under different fertilization treatments.

| Treatments | Exchangeable K (cmol $kg^{-1}$) | Exchangeable Na (cmol $kg^{-1}$) | Exchangeable Ca (cmol $kg^{-1}$) | Exchangeable Mg (cmol $kg^{-1}$) | EBC (cmol $kg^{-1}$) | ECEC (cmol $kg^{-1}$) |
|---|---|---|---|---|---|---|
| CK | 0.27 ± 0.05 a | 0.07 ± 0.01 a | 6.13 ± 0.38 a | 3.13 ± 0.47 a | 9.10 ± 0.87 ab | 9.37 ± 0.94 a |
| CF | 0.25 ± 0.02 a | 0.09 ± 0.01 a | 6.07 ± 0.22 a | 2.83 ± 0.19 a | 7.69 ± 0.40 b | 11.35 ± 0.99 a |
| OCF | 0.28 ± 0.04 a | 0.09 ± 0.02 a | 6.20 ± 0.32 a | 3.37 ± 0.28 a | 10.55 ± 0.45 a | 10.6 ± 0.70 a |
| OF | 0.31 ± 0.02 a | 0.11 ± 0.02 a | 6.70 ± 0.11 a | 3.80 ± 0.31 a | 10.92 ± 0.38 a | 10.17 ± 0.68 a |

Note: CK, no fertilization; CF, single application of chemical fertilizer; OCF, application of organic-inorganic fertilizer; OF, single application of organic fertilizer. EBC, total exchangeable base content; ECEC, effective cation exchange capacity. Different low case letters each line indicate significant differences ($p < 0.05$) among treatments using the least significant difference (LSD).

### 3.4. Soil $NO_3^-$-N, $NH_4^+$-N and NP

As shown in Table 4, the fertilization treatments could increase the contents of $NH_4^+$-N and $NO_3^-$-N. The $NH_4^+$-N content of the CF treatment was about four times ($p < 0.05$) those of the other treatments. Furthermore, the $NO_3^-$-N content of CF was also significantly higher than those of the OCF and CK treatments by 25.11% and 100.15%, respectively. The $NO_3^-$-N content of the OF treatment was significantly higher than CK by 85.30%. From the NP value obtained from the linear fitting equation, the NP values of the OCF, CK, and OF treatments varied from 0.04 to 0.09, while the CF treatment was significantly higher than the other treatments by 1.56–1.63 mg $kg^{-1}$ $h^{-1}$. Additionally, the NP value of OCF was the lowest among the four treatments.

**Table 4.** Concentration of $NH_4^+$-N and $NO_3^-$-N (0–20 cm) in tobacco growing field under different fertilizations.

| Treatments | $NH_4^+$-N (mg $kg^{-1}$) | $NO_3^-$-N (mg $kg^{-1}$) | NP Value (mg $kg^{-1}$ $h^{-1}$) | $R^2$ ($n = 4$) |
|---|---|---|---|---|
| | | | **NP** | |
| CK | 4.53 ± 0.82 b | 6.87 ± 0.41 c | 0.07 | 0.9996 |
| CF | 18.49 ± 1.74 a | 13.75 ± 0.95 a | 1.67 | 0.9989 |
| OCF | 4.97 ± 0.27 b | 10.99 ± 0.44 b | 0.04 | 0.9997 |
| OF | 4.59 ± 0.20 b | 12.73 ± 0.88 ab | 0.09 | 0.9754 |

Note: CK, no fertilization; CF, single application of chemical fertilizer; OCF, application of organic-inorganic fertilizer; OF, single application of organic fertilizer. $NH_4^+$-N, soil ammonium nitrogen; $NO_3^-$-N, soil nitrate nitrogen; NP, soil nitrification potential; $R^2$: the determination coefficient in the linear fitting equation. Different low case letters each line indicate significant differences ($p < 0.05$) among treatments using the least significant difference (LSD).

### 3.5. Soil Organic Carbon Pool

The effects of different fertilization treatments on the soil organic carbon pool and carbon sink management index (CPMI) were shown in Figure 3. With the increase in fertilization and cultivation years, each fertilization treatment increased the soil organic carbon pool and the labile organic carbon pool. Among the treatments, OCF and CF showed significant increases compared with CK, which were as high as 62.09% and 146.09% ($p <$ 0.05) for the soil organic carbon pool, and 38.70% and 97.71% ($p <$ 0.05) for the labile organic carbon pool, respectively. All the treatments increased CPMI compared with CK, among which the OCF and OF treatments had significantly increased by 162.69% and 108.51% ($p <$ 0.05), respectively.

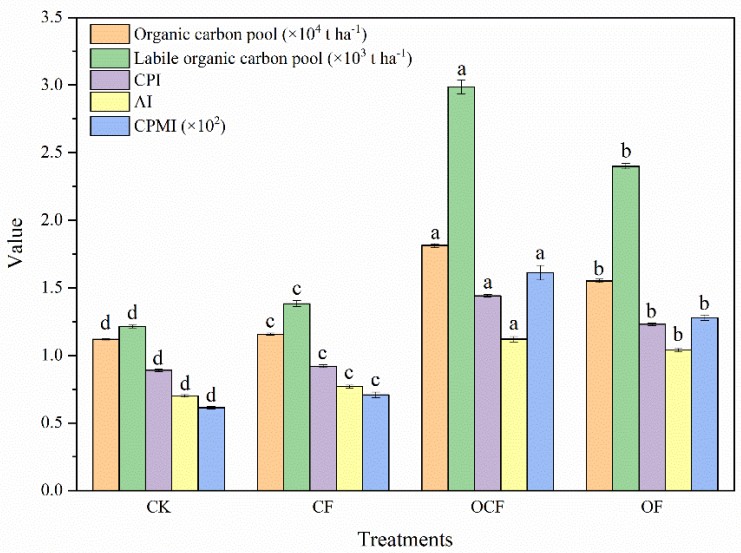

**Figure 3.** Soil organic carbon pool (0–20 cm) in tobacco growing field under different fertilizations. CK, no fertilization; CF, single application of chemical fertilizer; OCF, application of organic-inorganic fertilizer; OF, single application of organic fertilizer. CPI, carbon pool index; AI, activity index of the carbon pool; CPMI, carbon pool management index. Different low case letters each line indicate significant differences ($p <$ 0.05) among treatments using the least significant difference (LSD).

## 4. Discussion

### 4.1. Effects on Soil Acidification

It is suggested that balanced rates of N, P and K application, fertilization with reasonable amount and manners be taken to alleviate the problem of soil acidification, and to improve soil environmental quality. In this study, we found that the application of organic fertilizer for ten years significantly increased the soil pH, which was consistent with the research results of Rukshana et al. [29] and Tang et al. [30]. This confirmed the positive role of organic fertilizer in alleviating soil acidification. The following reasons can account for this phenomenon. Firstly, the decarboxylation of organic anions in organic fertilizer consumed the H$^+$ produced in the nitrification process of nitrogen fertilizer [29], which would increase the content of organic acid salt in the soil to decrease the pH. Secondly, organic fertilizer could reduce organic and inorganic nitrogen to slow the nitrification rate and reduce the production of nitrate nitrogen, and the range of this decrease increases with an increasing amount of application [31]. This could promote the absorption of inorganic nitrogen by tobacco and improve the utilization rate of nitrogen [32]. Thirdly, organic manure released organic matter into tobacco planting soil, increasing the chelation of exchangeable Al, and promoting the growth of tobacco [33]. In addition, there was a significant negative correlation between Ec and pH in this study. This was consistent with a well-known consequence that soil Ec is negatively correlated with pH in the acid soil and positively correlated with pH in the alkaline soil [34].

Compared with the soil active acidity represented by pH, exchangeable hydrogen and exchangeable Al represented the latent acid in soil, which was the acidity of $H^+$, $Al^{3+}$, or Al $(OH)^{2+}$ adsorbed on the surface of soil colloids after being exchanged by base [35]. In our ten-year positioning test, a significant difference observed among the four treatments was the lower exchangeable Al content in organic fertilizer treatment (Table 3). This was consistent with the results of Yu et al. [36] and Lu et al. [37], which confirmed that the application of organic fertilizer could reduce the soil pH by inhibiting the potential acid. Two reasons were responsible for this. On the one hand, the application of organic manure for many years would increase the content of soil organic matter, then, the organic matter would combine with exchangeable Al to form complex Al or organic-inorganic complexes which were not harmful to the soil [38]. On the other hand, organic manure could promote the hydrolysis of exchangeable Al in soil and transform it into Al hydroxy compound precipitates [39]. Figure 2 also confirmed that the decrease in exchangeable Al is the main factor in the decrease in exchangeable acid, which was related to the dominant position of exchangeable Al in the soil exchangeable acid. The release of exchangeable Al was closely related to soil parent material. Studies showed that the parent materials of brown soil were mostly residual slope deposits of granite, gneiss, sand shale and the order of exchangeable Al in soil developed from the various parent materials is shale > quaternary laterite > sandstone > limestone [40]. Therefore, the exchangeable acid of brown soil itself is mainly exchangeable Al. The significant changes of Al saturation with different treatments in this study also showed that the key to reducing the soil latent acid was to reduce the exchangeable Al in the soil, and the combined application of organic and inorganic fertilizers was the most effective method to achieve this.

*4.2. Effects on Soil Cation Exchange Capacity and Nitrification, and Their Relationships with Soil Acidification*

The effective cation exchange capacity (ECEC) includes the EBC, as well as exchangeable Al and hydrogen. Soil effective cation exchange capacity is also an important index for measuring the nutrient preserving capability of soil [41]. Therefore, an ideal management of fertilization should improve the soil effective cation exchange capacity, and its increase is mainly due to the increase in the exchangeable base, rather than the increase in exchangeable Al and hydrogen. Our long-term experiments indicated that although there was no significant difference in the ECEC among the four treatments, a significant improvement in the EBC with the application of organic fertilizer had been observed. In contrast, the use of chemical fertilizer alone played a significant role in reducing the EBC. This was consistent with many studies that inorganic nitrogen fertilizer could reduce the EBC in soil by promoting the absorption of base ions and nitrate leaching by crops [42], while organic fertilizer could increase the EBC in soil by decomposing and releasing base ions [43]. However, there were no significant effects on exchangeable K, Na, Ca, or Mg with the application of organic fertilizer. This contrasted with previous studies, which showed that the application of tobacco straw organic fertilizer significantly increased the EBC and the contents of exchangeable Na and Mg at the same time [44]. A reasonable explanation was that the kinds and quantities of base ions released from different organic fertilizers are different, and the contents of Na and Mg in cattle manure were lower [45].

Fertilization can affect soil acidification by affecting the process of the nitrogen cycle. Excessive fertilization can enhance nitrate leaching; the $H^+$ produced by nitrification cannot be neutralized and accumulated in the soil, resulting in acidification [46]. Our results showed that the application of chemical fertilizer leads to a large accumulation of $NH_4^+$-N, and the content of $NO_3^-$-N was significantly higher than other treatments, which increased the risk of soil acidification. This was related to the high ammonium nitrate ratio of the chemical fertilizer used in this study. Although the N content of OCF and OF treatment were higher than that of CF treatment (Table 1), most of them existed in the form of organic nitrogen and needed to be slowly transformed into available nitrogen nutrients such as $NH_4^+$-N and $NO_3^-$-N [47]. In addition, the C/N ratio of organic fertilizer was

relatively high, which promoted the nitrogen fixation process of microorganisms [48]. Therefore, the $NH_4^+$-N and $NO_3^-$-N of OCF and OF treatments were lower than that of CF. What is more, CF treatment resulted in a significant increase in soil NP compared with CK, OCF and OF treatments, indicating that the application of CF significantly increased soil nitrification. The relatively high content of $NH_4^+$-N in chemical fertilizer may be an important reason for accelerating the soil nitrification rate [49]. It could be seen that the use of chemical fertilizer to enhance soil nitrification was an important reason for the acidification of tobacco brown soil. However, combined application of organic fertilizer avoided large $NO_3^-$-N content and $H^+$ quantity in a short time by fixing inorganic nitrogen and weaken soil acidification caused by nitrification and $NO_3^-$-N leaching.

In order to further study the dominant role of different exchangeable base ions and nitrification in tobacco planting soil acidification, the redundancy analysis was used to rank the influencing factors (Figure 4). The results showed that NP (58.7%, $F = 19.9$, $p = 0.002$), $NH_4^+$-N (57.1%, $F = 18.6$, $p = 0.002$) and $NO_3^-$-N (22.5%, $F = 4.1$, $p = 0.036$) were important factors to promote soil acidification, while EBC (49.1%, $F = 13.5$, $p = 0.002$) and exchangeable K (24.6%, $F = 4.6$, $p = 0.016$) were important factors to inhibit soil acidification. Cations reduced soil acidity by exchanging with protons. The significant effect of exchangeable K was mainly due to the large amount of potassium fertilizer applied in the tobacco field, as many studies showed that NP was an important indicator of soil acidification, and had significantly positive relationship with soil acidification [50–52].

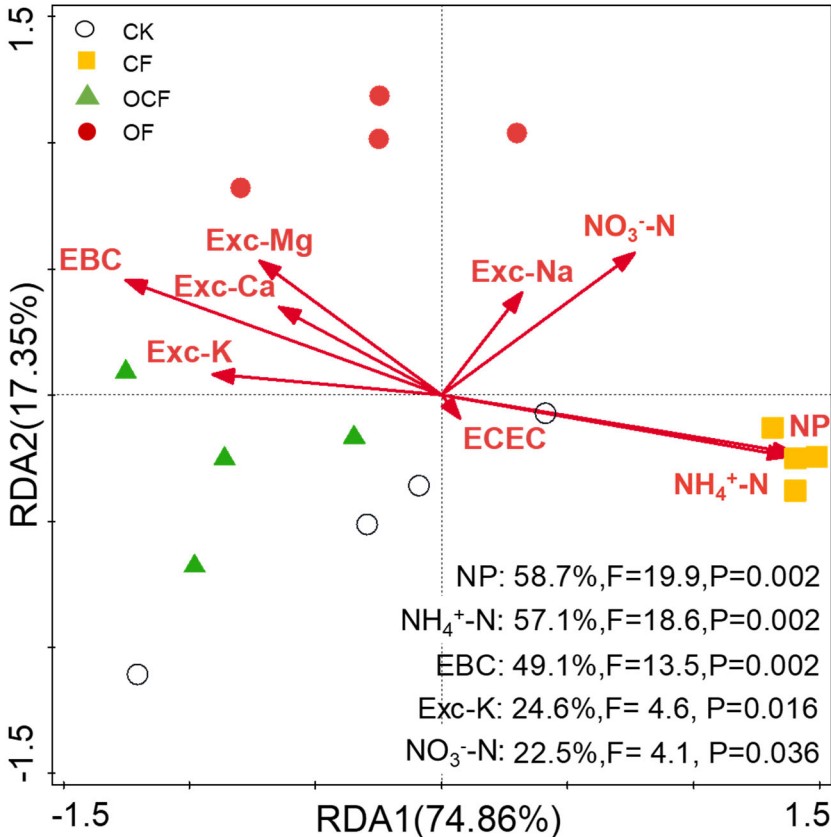

**Figure 4.** Redundancy analysis of soil acidification index and soil ions in tobacco planting brown soil. CK, no fertilization; CF, single application of chemical fertilizer; OCF, application of organic-inorganic fertilizer; OF, single application of organic fertilizer. EBC, total exchangeable base content; Exc-K, exchangeable potassium; Exc-Mg, exchangeable magnesium; Exc-Ca, exchangeable calcium; Exc-Na, exchangeable sodium; ECEC, effective cation exchange capacity; Exc-Aci, exchangeable acid; Exc-H, exchangeable hydrogen; Exc-Al, exchangeable aluminum; Al-Sat, Al saturation; Ec, electric conductivity.

### 4.3. Response and Causes of Organic Carbon Pool in the Process of Fertilization Regulating Soil Acidification

Organic material input played an important role in improving the soil carbon pool management index. Previous studies have confirmed that organic fertilizer input will inevitably cause dramatic changes in the soil carbon pool, which will lead to an increase in approximately 50% in CPMI with 5000 kg ha$^{-1}$ straw input for four years [28]. In our study, the application of organic fertilizer significantly increased the CPMI by 108.51%–162.69%, which was higher than the increase with crop straw. Cattle manure is a decayed fertilizer source which can provide soil with available nutrients directly and also promote the turnover of soil organic carbon. Thus, there was an increase in the CPI, AI, and CPMI in our study. In addition, whether using inorganic or organic fertilizer, either one would promote the increase of the CPMI. Because the available nutrients in chemical fertilizer release rapidly and promote the growth of tobacco roots and plants, resulting in more tobacco root residues remaining in the soil, it will increase the content of organic carbon in the soil after decomposition [53]. Compared with chemical fertilizer, adding organic fertilizer increased the CPI, AI, and CPMI more significantly. This was related to its advantages of being quick-acting and long-lasting, which could promote the growth and reproduction of soil microbes and the formation of diversity and enhance both soil water and fertilizer supply capacity [54,55]. Wang et al. [56] pointed out that pig manure combined with NPK increased C sequestration by 27.00%–64.40%, which was consistent with our result that organic-inorganic compound fertilizer most significantly increased the organic carbon pool by 62.09% (compared with no fertilizer), 56.60% (compared with only chemical fertilizer) and 16.86% (compared with only organic fertilizer). The significant increase in soil organic carbon had a direct effect on the decrease in acidogenic ions. The structural equation model (SEM) (Figure 5) further revealed the influence path of fertilization on organic carbon pool in the process of soil acidification. Soil nitrification, EBC, and ECEC could regulate soil organic carbon pool by affecting the change in soil acidity. Nitrification and EBC directly affected pH (SPC = −0.540, $p < 0.001$; SPC = 0.434, $p < 0.001$), and then indirectly affected CPMI (SPC = −0.018, $p < 0.05$). ECEC indirectly affected CPMI (SPC = 0.038, $p < 0.001$) and AI (SPC = −0.334, $p < 0.001$) by directly affecting exchangeable acidity (SPC = −0.302, $p < 0.01$) and exchangeable Al (SPC = −0.399, $p < 0.001$). This showed that reducing the nitrification caused by chemical fertilizer was the key to alleviate the soil acidification caused by long-term tobacco planting. At the same time, it was also necessary to increase the number of effective exchangeable cations to reduce the potential acidification threat caused by exchangeable Al. Organic fertilizer combined with chemical fertilizer alleviated the acidification of tobacco growing soil by increasing the number of exchangeable cations and reducing nitrification. Furthermore, it could significantly improve the management index of the soil carbon pool. Thus, organic fertilizer combined with chemical fertilizer was a feasible measure to reduce acid and increase carbon in tobacco planting soil.

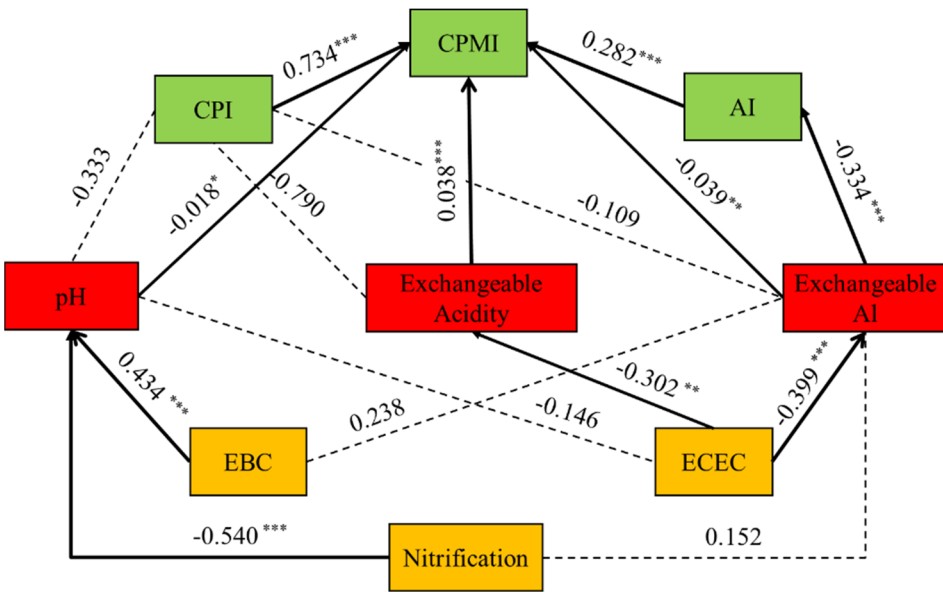

$\chi^2$=3.124; *df*=3; *P*=0.373; *GFI*=0.960; AIC=87.124; SRMR=0.003

**Figure 5.** Structural equation modeling (SEM) of the effects of submergence on organic carbon pool as a result of submergence-induced changes in soil acidification and soil environment. Square boxes denote variables included in the models. Values associated with solid arrows represent standard-ized path coefficients (SPCs) and asterisks mark their significance: *, Pb 0.05; **, Pb 0.01; ***, Pb 0.001. Solid arrows denote the directions and effects that were significant ($p < 0.05$), and the thickness rep-resents the magnitude of the path coefficients. Dashed arrows represent the directions and ef-fects that were non-significant ($p < 0.05$). EBC, total exchangeable base content; ECEC, effective cat-on exchange capacity; CPI, carbon pool index; AI, activity index of the carbon pool; CPMI, carbon pool management index.

## 5. Conclusions

The ten-year positioning experiment on a typical tobacco planting soil in China showed that the long-term application of OF and OCF could significantly alleviate the problem of soil acidification caused by the application of chemical fertilizer, which was manifested in increasing soil pH, reducing Ec and the exchangeable acid content (especially exchangeable Al). Compared with the application of chemical fertilizer, the appli-cation of organic fertilizer could significantly decrease soil nitrification and increase EBC. RDA showed that NP, $NH_4^+$-N, and $NO_3^-$-N were the important factors indicating soil acidification, while EBC and exchangeable K were the significant factors restricting soil acidification. In addition, adding organic fertilizer had a remarkable increase on the or-ganic carbon pool and labile organic carbon pool. The structural equation model (SEM) showed that OCF treatment increased the soil organic carbon pool mainly by inhibiting soil nitrification and reducing the content of exchangeable Al. In conclusion, both OF and OCF treatments were effective methods to alleviate tobacco planting soil acidification, but OCF had more advantages in improving the soil organic carbon pool.

**Author Contributions:** Conceptualization, P.D.; methodology, P.C. and W.S.; validation, P.D.; for-mal analysis, P.W. and P.C.; investigation, J.D.; data curation, W.S. and Z.D.; writing—original draft preparation, P.D. and P.C.; writing—review and editing, P.C. and P.W.; supervision, P.D.; project administration, Z.D.; funding acquisition, J.D. All authors have read and agreed to the published version of the manuscript.

**Funding:** This work was funded by the Central Public-interest Scientific Institution Basal Research Fund (1610232020009), the Science Foundation for Young Scholars of Tobacco Research Institute of Chinese Academy of Agricultural Sciences (No.2020B01), the Yunnan Branch of China National To-bacco Corporation, China (2019530000241017) and the Provincial Natural Science Foundation of An-hui (No.2108085QC112).

**Institutional Review Board Statement:** Not applicable.

**Informed Consent Statement:** Not applicable.

**Data Availability Statement:** The data presented in this study are available on request from the corresponding author.

**Conflicts of Interest:** The authors declare no conflict of interest.

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
