# Peer review of "Alleviating Soil Acidification and Increasing the Organic Carbon Pool by Long-Term Organic Fertilizer on Tobacco Planting Soil"

_agronomy, doi:10.3390/agronomy11112135_

Round 1
Reviewer 1 Report
The manuscript "Alleviating soil acidification and increasing the organic carbon pool by long-term organic fertilizer on tobacco planting soil" deals with an interesting subject, but some minor revisions are necessary!
My most serious criticism of the manuscript relates to the premise of the questions. There is no scientific hypothesis! Formulated aims of the paper are not scientific questions. So I recommend reformulating the last paragraph of the introduction.
The paper is fundamentally good. I have only some minor comments:
- Application of superscript in case of ammonium and nitrate nitrogen. It looks like the + and - are subscript, not superscript.
- Fig 1:b The ±0,1-0,2 pH is not a difference between measurements. I think that the Authors have to rethink these results.
Author Response
Point 1: My most serious criticism of the manuscript relates to the premise of the questions. There is no scientific hypothesis! Formulated aims of the paper are not scientific questions. So I recommend reformulating the last paragraph of the introduction.
Response 1: The last paragraph of the introduction had been reformulated. The scientific question we raised is what is the way for long-term application of organic fertilizer to alleviate soil acidification. Please see L.78-L.87 for details.
Point 2: Application of superscript in case of ammonium and nitrate nitrogen. It looks like the + and - are subscript, not superscript.
Response 2: We have revised the superscript and subscript of the symbols in the text. It seems that there is ambiguity, which may be due to the font and display.
Point 3: Fig 1:b The ±0,1-0,2 pH is not a difference between measurements. I think that the Authors have to rethink these results.
Response 3: Fig. 1b is a significance analysis between the pH units increased or decreased by each treatment compared with CK. It is the difference between pH changes, not the difference between original pH. We have rechecked the data.
Reviewer 2 Report
The manuscript by Dai et al. reports on an important issue concerning long-term application of mineral fertilizers and their negative outcomes for agriculture, soil productivity and environmental sustainability. Overall this MS is very well organized. I recommend major revision. Checking overall English expressions would be recommended.
Here are my suggestions and comments
- The first sentence of abstract need to be rephrased. Incorrect grammatically
- Line 29, seems something missing (both of and OCF)??
- What do authors mean by flue-cured tobacco?
- Line 43, I wonder if nicotine is also a heavy metal
- Line 62, provide a suitable reference
- Line 75, re-order the citations as 8,14, 18.
- The introduction section provides sufficient information on study background and highlights the problem well. However I suggest authors to enrich citation with more recent references. Some are provided here.
https://doi.org/10.1016/j.jenvman.2020.110894
https://doi.org/10.1016/j.scitotenv.2020.138181
https://doi.org/10.1016/j.apsoil.2021.103971
https://doi.org/10.1007/s11368-021-03048-0
- Line 101, long-term positioning test?
- Line 106, double check the temperature range
- How the animal manure was applied? Do all the fertilizers applied each year? What does single chemical fertilizer means?
- Provide the commercial providers information for fertilizers used.
- Do the authors adjusted the rate of N in organic fertilization plots?especially in organic plus chemical fertilizer treatment?
- I suggest authors to double check if the rates provided in Table 1 are for P2O5 and K2O or P or K.
- Line 139, mature season??
- Was there only one replicate for each treatment?
- Line 188, the TOC content given is 5.26 g/kg while in line 11 it is 11.66 g/kg. why there are 2 values? What is the difference?
- Lines 181-186, the calculations for Carbon management index are not correct. First, it is Carbon management index (CMI) and not the carbon pool management index (CPMI). Second it is calculated by a multiple of CPI and LI (lability index) and not AI (activity index) which you are using here. Please justify and correct it in the manuscript.
- Line 208-209, replace was shown to have been shown. Check other places too.
- Provide in figure 2 the purpose of dark and light colors.
- Table 4, superscript R2
- I suggest converting data in tabe 5 to figures after correction of CMI and LI
- Please check the calculation for organic carbon pool and labile organic carbon pool. They are too high in table 5.
- Lines 324 -325 relate with your findings
- Discussion section is quite good however the discussion on mechanisms should be explored deeper.
- Conclusion is fine and supported by the findings.
- About 95% references are from Chinese scientists. Add some references from the rest of world to give a universal value.
Author Response
Dear Editor and reviewer,
On behalf of my co-authors, we thank you very much for giving us an opportunity to revise our manuscript, we appreciate reviewers very much for their positive and constructive comments and suggestions on our manuscript entitled“Alleviating soil acidification and increasing the organic carbon pool by long-term organic fertilizer on tobacco planting soil”.
We have studied reviewer’s comments carefully and have tried our best to revise our manuscript according to the comments.
Response to Reviewer 2 Comments
Point 1: The first sentence of abstract need to be rephrased. Incorrect grammatically
Response 1: We have rephrased the first sentence of abstract with “Long term tobacco planting leads to soil acidification”.
Point 2: Line 29, seems something missing (both of and OCF)??
Response 2: Yes, Of should be capitalized. It is “OF and OCF”, not “of and OCF”. We have corrected it.
Point 3: What do authors mean by flue-cured tobacco?
Response 3: We have revised flue-cured tobacco with Nicotiana tabacum.
Point 4: Line 43, I wonder if nicotine is also a heavy metal
Response 4: Thanks for your advice. Nicotine is not a heavy metal. We have deleted it. Please see L.42.
Point 5: Line 62, provide a suitable reference
Response 5: We have added the following reference in L.62.
Lv M.R., Li Z.P., Che Y.P., Han F.X., Liu M. Soil organic C, nutrients, microbial biomass, and grain yield of rice (Oryza sativa L.) after 18 years of fertilizer application to an infertile paddy soil. Biol. Fertil. Soils, 2011, 47, 777–783.
Point 6: Line 75, re-order the citations as 8,14, 18.
Response 6: We have re-ordered the citations as 8, 20. All reference numbers have also been adjusted because two references were added.
Point 7: The introduction section provides sufficient information on study background and highlights the problem well. However, I suggest authors to enrich citation with more recent references. Some are provided here.
Response 7: Thanks for your latest literature. We have replaced it with the more recent references in this paper. Please see L.61 and L.64. The number of the references is [14], [15], [16], and [17].
Point 8: Line 101, long-term positioning test?
Response 8: We have replaced it with “long-term field experiment”.
Point 9: Line 106, double check the temperature range
Response 9: We have checked that the annual average temperature is 12.1℃ and the annual accumulated temperature is 4410 ℃.
Point 10: How the animal manure was applied? Do all the fertilizers applied each year? What does single chemical fertilizer means?
Response 10: “All fertilizers were applied to the surface in a certain amount, and then mixing with 0-10 cm soil manually, ridging and planting tobacco. The fertilizers were applied before tobacco planting every year. ” We have added it in L.128 to L.130.
Single chemical fertilizer means applying chemical fertilizer only.
Point 11: Provide the commercial providers information for fertilizers used.
Response 11: We have added the commercial providers of the fertilizers. Please see L.121-L.123.
Point 12: Do the authors adjusted the rate of N in organic fertilization plots? especially in organic plus chemical fertilizer treatment?
Response 12: Yes, compared with CF, OCF treatment is to increase the application of rotten cow dung as an organic nitrogen source under the condition of constant inorganic nitrogen application, and study the mechanism of adding the organic fertilizer to alleviate soil acidification and increase the organic carbon pool of tobacco planting soil.
Point 13: I suggest authors to double check if the rates provided in Table 1 are for P2O5 and K2O or P or K.
Response 13: We have checked that the rates provided in Table 1 are for P2O5 and K2O.
Point 14: Line 139, mature season??
Response 14: We have corrected with “mature period”.
Point 15: Was there only one replicate for each treatment?
Response 15: No. The experiment used a randomized block design with three replicates. We have informed this in L.119.
Point 16: Line 188, the TOC content given is 5.26 g/kg while in line 11 it is 11.66 g/kg. why there are 2 values? What is the difference?
Response 17: The 11.66 g/kg is the SOM content of the tested soil at the beginning of the experiment (in the year of 2010), while 5.26 g/kg is SOC content of abandoned land soil in 2020.
Point 17: Lines 181-186, the calculations for Carbon management index are not correct. First, it is Carbon management index (CMI) and not the carbon pool management index (CPMI). Second it is calculated by a multiple of CPI and LI (lability index) and not AI (activity index) which you are using here. Please justify and correct it in the manuscript.
Response 17: Thanks for your advice. The calculations of CPMI and AI referred to the reference [27] and [28]. Because we calculated the soil organic carbon pool in Table 5, the soil carbon pool management index was used here. The references showed that AI also could be used to calculate CPMI. And it has the same meaning as LI in here.
[27] Blair G.J., Lefroy, R.D., Lisle L. Soil carbon fractions based on their degree of oxidation, and the development of a carbon management index for agricultural systems. Crop Pasture Sci., 1995, 46, 1459–1466.
[28] Wang X.H., Yang H.S., Liu J., Wu J.S., Chen W.P., Wu J., Zhu L.Q., Bian X.M. Effects of ditch-buried straw return on soil or-ganic carbon and rice yields in a rice—wheat rotation system. Catena, 2015, 127, 56–63.
Point 18: Line 208-209, replace was shown to have been shown. Check other places too.
Response 18: We have replaced it with “have been shown”. Please see L.206.
Point 19: Provide in figure 2 the purpose of dark and light colors.
Response 19: The dark and light colors in Fig.2 means the strong and weak correlations. We have added it in the note.
Point 20: Table 4, superscript R2
Response 20: We have superscript R2.
Point 21: I suggest converting data in table 5 to figures after correction of CMI and LI
Response 21: We have converted data in table 5 to figure. Please see figure 3.
Point 22: Please check the calculation for organic carbon pool and labile organic carbon pool. They are too high in table 5.
Response 22: Table 5 calculates the organic carbon storage of 0-20 cm soil layer, not topsoil. We have calculated it again and confirm the consequence.
Point 23: Lines 324 -325 relate with your findings
Response 23: In addition, there was a significant negative correlation between Ec and pH in our study. This was consistent with a well-known consequence that soil Ec is negatively correlated with pH in the acid soil and positively correlated with pH in the alkaline soil [34].
We have revised it as above. Please see L.322-L.323.
Point 24: Discussion section is quite good however the discussion on mechanisms should be explored deeper.
Response 24: We have added some of them in-depth discussions. Please see L.358-L.362 and L.380-L.383.
Point 25: Conclusion is fine and supported by the findings.
Response 25: Thank you!
Point 26: About 95% references are from Chinese scientists. Add some references from the rest of world to give a universal value.
Response 26: Thanks for your advice. We have revisited some references and replaced them with other foreign references.
We would like to express our great appreciation to you for comments on our paper. Looking forward to hearing from you.
Thank you and best regards.
Yours sincerely,
With kindly regards

This manuscript is a resubmission of an earlier submission. The following is a list of the peer review reports and author responses from that submission.